# Anxiety and depression in Guatemala: Sociodemographic characteristics and service access

**Jonathan Naber** [1]*, **Islay Mactaggart**[1], **Carlos Dionicio**[2], **Sarah Polack**[1]

**1** International Centre for Evidence in Disability, London School of Hygiene & Tropical Medicine, London, United Kingdom, **2** Consejo Nacional para la Atención de las Personas con Discapacidad (CONADI), Guatemala City, Guatemala

* jonathan@rompglobal.org

**Data Availability Statement:** The data underlying this study cannot be made publicly available, as approval was not obtained from study participants to make their data openly available due to legal and ethical restrictions imposed by the national board

## Abstract

Epidemiological data on depression and anxiety in Guatemala is lacking. Using 2016 National Disability Survey data, we explored the sociodemographics of people with anxiety and/or depression and its heightened burden on access to key services. The survey (n = 13,073) used the Washington Group Extended Set to estimate disability prevalence, including anxiety and/or depression. A nested case-control study was included to explore the impact of disability on key life areas. Cases (indicating 'A lot of difficulty' or 'Cannot do' in one or more functional domain) and age-/sex-matched controls were administered a structured questionnaire. Multivariable logistic regression and heightened-burden analysis were conducted. Higher odds of anxiety and/or depression were found in participants who were 50+ (aOR 2.3, 1.8–3.1), female (aOR 1.8, 1.4–2.2), urban (aOR 1.5, 1.2–1.9), divorced/separated (aOR 2.0, 1.3–3.0), and widowed (aOR 1.6, 1.0–2.4), as well as those with impaired communication or cognition (aOR 17.6, 13.0–23.8), self-care (aOR 13.2, 8.5–20.5), walking (aOR 13.3, 9.7–18.3), hearing (aOR 8.5, 5.6–13.1), and vision (aOR 8.5, 6.1–11.8). Lower odds of anxiety and/or depression were found in participants with a university education (aOR 0.2, 0.5–0.9), and those living in the southeast (aOR 0.2, 0.1–0.3) or northeast (aOR 0.3, 0.2–0.4). Compared to people with impairments that were not depression and/or anxiety, people with depression and/or anxiety were less likely to receive a retirement pension (aOR 0.4, 0.2–0.8), and more likely to receive medication for depression/anxiety (aOR 4.1, 1.9–9.1), report a serious health problem (aOR 1.8, 1.3–2.5), and seek advice/treatment with a government health worker/health post (aOR 6.3, 1.0–39.2).

## Introduction

According to a 2017 report by the World Health Organization (WHO), 4.4% of the global population had a depressive disorder in 2015—a total of 322 million people—and 3.6% had an anxiety disorder—a total of 264 million people. Depressive disorders, which include major depression and dysthymia, are the leading cause of Years Lived with Disability (YLDs) both

of Guatemala. Researchers interested in the data may contact Islay Mactaggart at the Guatemala National Study (email: islay.mactaggart@lshtm.ac. uk) or the International Centre for Evidence in Disability (email: disabilitycentre@lshtm.ac.uk) for data access.

**Funding:** The Guatemala survey was funded by CBM Latin America, CONADI (the National Council on Disability), and UNICEF, in grants to Dr. Sarah Polack. No specific funding was received for these analyses of the data.

**Competing interests:** The authors have declared that no competing interests exist.

globally and in the Region of the Americas. Anxiety disorders, which include a number of conditions, are close behind as the sixth leading cause of YLDs globally and the third leading cause of YLDs in the Region of the Americas [1].

Guatemala created a National Mental Health Policy in 2007, defining its strategic lines and roles within the public health system for addressing the mental health needs of the population [2]. In its most recent evaluation of the Guatemalan mental health system, in 2011, the WHO reported that the 2007 policy had not yet passed through the necessary ministerial and legislative processes to be implemented [3–5]. This has perpetuated the extremely low proportion of funding earmarked for mental health—stagnated since 2007 at less than 1% of public health spending—as well as the scarcity of mental health practitioners, the centralization of services in urban areas, and the exclusion of rural and indigenous populations [3–6]. Kohn et al. estimated that the treatment gap of anxiety disorders and affective disorders—that is, the prevalence of people with these disorders not receiving mental health treatment in the past 12 months—is 95.1% and 97.1%, respectively, in Guatemala [7].

There is a lack of epidemiological data on mental illnesses in Guatemala [2, 4, 5, 8]. The only national, population-representative data comes from the 2009 National Survey on Mental Health (ENSM), which measured the prevalence of mental health disorders in 1,452 participants throughout the country via the Composite International Diagnostic Interview (CIDI) Version 2.1 [4]. The CIDI instrument is based on the combined diagnostic systems of the Diagnostic and Statistical Manual (DSM) and the International Classification of Diseases (ICD), and measures a wide range of substance abuse, psychotic, mood, and anxiety/somatoform disorders [4, 5, 9]. The ENSM study found the overall prevalence of mental health disorders to be 27.8% in the surveyed population, with anxiety/somatoform and mood disorders most prevalent, at 20.2% and 8.0%, respectively. This study generated important data for Guatemala. However, it only included adults between the ages of 18 and 65 effectively excluding half the total population [4, 10]. A similar study was conducted in the Guatemala City metropolitan region in 2011, using the CIDI Version 3.0. This study found the overall prevalence of mental health disorders to be 29.0%, with anxiety and substance abuse disorders most prevalent, at 14.0% and 11.3%, respectively [5]. Other studies used service-based sampling and were not representative of the general population [11–14].

The 2016 Guatemalan National Disability Survey (ENDIS) was a national, population-representative survey that measured the prevalence of disability in 13,073 participants throughout Guatemala via the Washington Group Extended Set on functioning (WG-ES, adults 18+) and the WG-UNICEF Child Functioning Module (CFM, children 2–17) [15]. These instruments are based on the International Classification of Functioning, Disability, and Health (ICF) and widely used in population-based surveys [16]. They measure self-reported functioning by asking about level of difficulty with basic universal activities. They also include questions on psychosocial function, asking about frequency and intensity of feelings of anxiety and depression [16–18]. The study found the prevalence of anxiety and/or depression was 11.8% in women and 5.9% in men, while in girls 5–17 it was 2.8% and in boys 5–17 2.1%. The survey included a nested case-control study comparing cases (people with disabilities) with age- and sex-matched controls to determine associations between disability and a variety of indicators including health conditions and service access [15].

The substantial gap in mental health spending and treatment in Guatemala suggests that people with anxiety and depression may face heightened burdens compared to people with other disabilities or with no disabilities. Recent studies by MacTaggart et al compared outcomes of people with a certain disability type to people with/without other types of disabilities. This allowed for the identification of excess risk of various outcomes associated with the disability of focus [19, 20]. The objectives of our paper are to study 1) the sociodemographic

characteristics of people with anxiety and/or depression, and 2) the additional impact of having symptoms of anxiety/depression on health conditions, healthcare access, education, and livelihood, compared to people with other types of impairments.

## Methods

### Design and sampling

ENDIS was a national, population-representative survey of disability prevalence conducted in 2016, by the Guatemalan National Disability Council (CONADI) in partnership with the London School of Hygiene & Tropical Medicine (LSHTM), CBM, and UNICEF. Each of the 22 departments (provinces) of Guatemala was assigned to one of five geographical regions of the country. A sample size of 2,760 people were to be surveyed in each of the five regions to measure an estimated 6% all-age prevalence of disability with 95% confidence and 20% precision, assuming a 1.5 design effect, and 15% rate of non-response. The sample size for the entire country was therefore 13,800 (280 clusters of 50 people). Participants were selected through multi-stage stratified cluster random sampling with probability proportional to size. In the first stage of sampling, each region (Central, Northeast, Northwest, Southeast, and Southwest) was stratified on rural/urban designation, and a total of 56 census sectors of approximately 1,000 people were then selected from each region's strata. This was conducted by the National Institute of Statistics (INE) using the 2009 Guatemalan National Census as the sampling frame. In the second stage of sampling, compact segment sampling (CSS) was used to randomly select one segment containing approximately 50 people from each cluster, approximately 10 households in total. This stage of sampling was conducted by field staff using maps produced through GIS software. Nine of the originally-selected clusters were replaced due to lack of permission from community leaders or security concerns.

### Case definition and data collection

All members of each household aged 2+ years were interviewed using the WG instruments (WG-ES for 18+ and CFM for 2–17) [21]. The WG ES was developed, tested and adopted by the Washington Group on Disability Statistics. The CFM was developed and tested by the WG together with UNICEF. They are both internationally recognised tools widely used in surveys to collect comparable data on disability [22]. The question sets have undergone cognitive and field testing in different settings. Additionally studies of the CFM have found good internal- and factor-level consistency and substantial or moderate inter-rater and test-retest reliability, although similar studies for the WG-ES are lacking [23, 24].

These question sets assess self-reported functioning by asking about level of difficulty (none, some, a lot or cannot do) with basic universal activities in the domains of seeing, hearing, walking, self-care, cognition, communication, upper body activities, and self-care. In ENDIS, in line with Washington Group recommendations, anyone reporting 'A lot of difficulty' or 'Cannot do' in one or more of these functional domains was considered to have a disability. Additionally, the WG-ES and CFM include questions on frequency and intensity of feelings of anxiety and depression (see Table 1). Table 1 shows case definitions used in this study to indicate likely anxiety and depression, following recommendation of the tool developers and as used in previous studies [15, 25]. These questions are designed to be relatively quick and simple to use in surveys in different contexts. Analysis assessing the relationship of these questions to mental health screening tools K6 and PHQ-9, suggest some evidence for convergent construct validity [26].

For children aged 5–10 years, parents/guardians were interviewed as proxies. Participants aged 11 and older were interviewed directly.

**Table 1. Questions and case definitions for anxiety and depression.**

| | Responses Classified as having anxiety/ depression | Responses Not Classified as having anxiety/depression |
|---|---|---|
| **Age 5–17** | | |
| How often does [name] seem anxious, nervous or worried? | Daily (to one or both questions) | Weekly, Monthly, A Few Times a Year, Never |
| How often does [name] seem sad or depressed? | | |
| **Age 18+** | | |
| How often do you feel worried, nervous or anxious? | Daily (to first) AND A Lot (to second) | Weekly, Monthly, A Few Times a Year, Never (to first) AND/OR None or Some (to second) |
| Thinking about the last time you felt worried, nervous or anxious, how would you describe the level of these feelings? | | |
| How often do you feel depressed? | Daily (to first) AND A Lot (to second) | Weekly, Monthly, A Few Times a Year, Never (to first) AND/OR None or Some (to second) |
| Thinking about the last time you felt depressed, how depressed did you feel? | | |

## Co-variates

For all survey participants, data were collected on individual (age, sex, ethnic group, education, literacy and marital status) and household (rural/urban, region and household building materials and ownership of assets) level socio-economic characteristics.

In addition, a nested case-control study was conducted. This included all people identified as having a disability ('cases') according to the study definition (i.e. reported a 'lot of difficulty' or 'cannot do' in at least one functional domain or 'daily' and 'a lot' to the anxiety/depression questions) and, for each case, one age-sex and cluster- matched control without a disability. Age matching allowed for +/- 2 years for child cases and +/- 10 years for adult cases. Cases and controls were interviewed using a standardised questionnaire which included questions on current school attendance (children <18 years) and current work status and receipt of social protection (adults 18+ years). Additionally, for a list of 20 health conditions, participants were asked whether or not they had been diagnosed with this condition by a health professional and, if yes, whether they had received medication for that condition in the past 12 months. Other data on health included experience of serious health conditions in the past 12 months, type of health service utilised and experiences of health care (level of respect from health professionals, ease of understanding information/being understood).

## Translation and app

All data collection instruments were translated into Guatemalan Spanish and then back-translated into English to ensure original concepts were retained. The WG instrument was forward translated from Spanish into the four leading non-Spanish languages of Guatemala—K'iche', Kaqchikel, Mam, and Q'eqchi' and then back-translated to check concepts. The questionnaire was pilot tested with up to five people in each language to assess comprehension and equivalence, with adaptations made accordingly. Field staff were recruited who spoke one of these languages. Local interpreters were also recruited within the clusters being surveyed, particularly in the Northwestern region where many people's primary language is a non-Spanish language. A tablet-based application was developed for all data collection, including the household roster, WG screens, clinical screens, and case-control questionnaires. Data were collected on tablets and uploaded to a secure, cloud-based server on a daily basis during the survey.

## Team, training, and field operations

The field staff included a survey coordinator, assistant, supervisor, and 18 interviewers. A five-day training was given which covered all topics relevant to field work. Five survey teams were created, each of which was responsible for surveying approximately 11 clusters per region. The survey coordinator, assistant, and supervisor regularly accompanied the interviewers to conduct quality control. Permission to conduct the survey was obtained from local authorities from each of the clusters prior to the start of data collection, either municipal or indigenous leaders. Information about the survey was spread through national radio, television, and newspapers to inform the population of the survey and improve trust of the surveyors and thus response rates. Police escorts were arranged for surveying in insecure clusters, primarily in the Guatemala City metropolitan area. A national directory of disability services was created in partnership with ASCATED, CBM, and CONADI, and distributed to the public health center closest to each surveyed cluster to facilitate the referral process of people identified with disabilities/functional difficulties. At the end of the survey when all clusters had been visited, an additional two weeks was spent revisiting clusters with low response rates or incomplete data (primarily in the Central and Northwest regions) to maximise response rate.

## Ethics

ENDIS received ethical approval by the observational ethics review committee of the London School of Hygiene & Tropical Medicine (LSHTM) in the United Kingdom, and the ethics review committee Latin Ethics in Guatemala. Information about the study was provided verbally to all participants and participants were also given the information sheet to read themselves. Written informed consent was obtained from all participants, and in the case of people aged < 18 years it was taken from their adult guardian.

## Data analysis

Data were analysed in STATA 16. We created the combined 'anxiety/and or depression' variable for the purposes of comparing to people reporting severe difficulties in other functional domains (e.g. sensory, mobility, self-care communication, cognition), but not mental health. Principal component analysis (PCA) was undertaken to construct household-level Socio-economic Position (SEP) scores based on household characteristics and ownership of durable assets. Multivariable logistic regression was conducted to i) compare sociodemographic differences between people in the full survey with and without anxiety and /or depression and ii) between 'cases' (included in the Case Control study) with anxiety and/or depression and 'cases' with other functional difficulties, but not anxiety/depression. All regression analysis were adjusted for age, sex, region and socio-economic position (SEP) as potential confounders.

## Results

A total of 13,073 people were included in the study (response rate of 95%). The age and sex distribution of the study population were broadly similar to that of the national population (based on the 2019 census, Table 2). Therefore, reported prevalence estimates are self-weighting.

A total of 385 people aged 5+ reported anxiety and /or depression according to the case definition (3.0%, 2.6–3.4).

**Table 2. Age and sex distribution of the national population and study sample.**

| | Male | | Female | | Total | |
|---|---|---|---|---|---|---|
| Age group | National | Sample | National | Sample | National | Sample |
| 0–14 years | 2,810,621 (34%) | 2216 (37%) | 2,699,230 (32%) | 2220 (31%) | 5,509,851 (33%) | 4,146 (34%) |
| 15–24 years | 1,753,082 (21%) | 1323 (22%) | 1,720,392 (20%) | 1582 (22%) | 3,473,474 (21%) | 2,905 (22%) |
| 25–54 years | 2,847,303 (35%) | 1772 (29%) | 3,095,531 (37%) | 2435 (35%) | 5,942,834 (36%) | 4,208 (32%) |
| 55–64 years | 368,242 (5%) | 325 (5%) | 447,825 (5%) | 413 (6%) | 816,067 (5%) | 738 (6%) |
| 65+ years | 395,467 (5%) | 397 (6%) | 466,333 (6%) | 409 (6%) | 861,800 (5%) | 806 (6%) |
| Total | 8,174,715 (49%) | 6033 (46%) | 8,429,311 (51%) | 7,039 (54%) | 16,604,026 | 13,073 |

*Data on sex missing for 1 person; Source of national estimates: Guatemala National Institute of Statistics 2019 [10].

## Sociodemographic characteristics

Table 3 compares the socio-demographic characteristics of people with anxiety and /or depression (n = 385) and without (n = 12,688, full survey sample). People in the 50+ range were over twice as likely to experience anxiety and/or depression compared to those in the 5–17 age range (aOR 2.3, 1.8–3.1). Females were nearly twice as likely to experience anxiety and/or depression compared to males (aOR 1.8, 1.4–2.2). People living in the Southeast (aOR 0.2, 0.1–0.3) and Northeast (aOR 0.3, 0.2–0.4) regions had significantly lower odds of anxiety and/or depression t compared to those in the Central region. Living in an urban area was also associated with significantly increased risk of anxiety and/or depression compared to living in a rural area (aOR 1.5, 1.2–1.9). People with university level of education were much less likely to report anxiety and/or depression compared to those with no formal education (aOR 0.2, 0.5–0.9). People who were divorced/separated (aOR 2.0, 1.3–3.0) and those who were widowed (aOR 1.6, 1.0–2.4) were significantly more likely to experience anxiety and/or depression compared to people who were married/living together. Ethnic group, socioeconomic position, and literacy were not significantly associated with anxiety and/or depression.

We also investigated whether having functional limitations in other WG domains was associated with anxiety and/or depression. Reporting a functional limitation in any of the other WG domains was associated with a substantially higher odds of anxiety and/or depression: Communication or cognition limitation (aOR 17.6, 13.0–23.8), Self-care (aOR 13.2, 8.5–20.5), Walking (aOR 13.3, 9.7–18.3), Hearing (aOR 8.5, 5.6–13.1), and Vision (aOR 8.5, 6.1–11.8).

## Education and livelihood

Table 4 compares education and livelihood between 'cases' with anxiety and/or depression (n = 385) and cases without anxiety and/or depression (n = 548, other people with disabilities). For people in the 5–17 age range, being currently enrolled/not in school was not associated with anxiety and/or depression. Neither was working/not in the past seven days for people in the 18+ age range. However, over half of the people in each group had not worked in the past seven days, at 62% and 61%, respectively. In terms of state benefits, cases with anxiety and/or depression had significantly lower odds of receiving a retirement pension compared to cases without anxiety and/or depression (aOR 0.4, 0.2–0.8).

## Health conditions

Table 5 compares access to medications between cases with and without anxiety and/or depression. The table shows medication for health conditions diagnosed by a doctor in both groups. A wide range of health conditions were included due to the wide-ranging impact of

**Table 3. Socio-demographic characteristics of people with and without anxiety and/or depression (n = 13,073).**

| | People with anxiety and/or depression (n = 385) | People without anxiety and/or depression (n = 12,688) | Unadjusted OR (95% CI) | Age, sex, adjusted OR (95% CI) |
|---|---|---|---|---|
| | N (%) | N (%) | | |
| **Age** | | | | |
| 5–17 | 107 (28%) | 4,293 (34%) | Baseline[‡‡] | Baseline[‡‡] |
| 18–49 | 164 (43%) | 5,405 (43%) | 1.2 (0.9–1.6) | 1.2 (0.9–1.5) |
| 50+ | 114 (30%) | 1,921 (15%) | 2.4 (1.8–3.1)[‡‡] | 2.3 (1.8–3.1)[‡‡] |
| **Sex** | | | | |
| Male | 124 (32%) | 5,909 (47%) | Baseline | Baseline |
| Female | 261 (68%) | 6,778 (53%) | 1.8 (1.4–2.2)[‡‡] | 1.8 (1.4–2.2)[‡‡] |
| **Region** | | | | |
| Central | 120 (31%) | 1,935 (17%) | Baseline | Baseline |
| Northeast | 37 (10%) | 2,445 (21%) | 0.2 (0.2–0.4)[‡‡] | 0.3 (0.2–0.4)[‡‡] |
| Northwest | 124 (32%) | 2,277 (20%) | 0.9 (0.7–1.1) | 0.9 (0.7–1.2) |
| Southeast | 26 (7%) | 2,581 (22%) | 0.2 (0.1–0.3)[‡‡] | 0.2 (0.1–0.3)[‡‡] |
| Southwest | 78 (20%) | 2,381 (20%) | 0.5 (0.4–0.7)[‡‡] | 0.5 (0.4–0.7)[‡‡] |
| **Location** | | | | |
| Rural | 193 (50%) | 7,134 (61%) | Baseline | Baseline |
| Urban | 192 (50%) | 4,485 (39%) | 1.5 (1.3–1.9)[‡‡] | 1.5 (1.2–1.9)[‡‡] |
| **Ethnic Group** | | | | |
| Maya | 180 (47%) | 5,265 (45%) | Baseline | Baseline |
| Latino/Mix | 194 (50%) | 5,820 (50%) | 1.0 (0.8–1.1) | 0.9 (0.8–1.1) |
| Other | 3 (1%) | 131 (1%) | 0.7 (0.2–2.1) | 0.7 (0.2–2.3) |
| Not Specified | 8 (2%) | 403 (4%) | 0.6 (0.3–1.2) | 0.6 (0.3–1.2) |
| **SEP** | | | | |
| 1st Quartile (poorest) | 75 (19%) | 2,942 (25%) | Baseline | Baseline |
| 2nd Quartile | 98 (25%) | 2,989 (26%) | 1.3 (0.9–1.7) | 1.3 (0.9–1.7) |
| 3rd Quartile | 110 (29%) | 2,919 (25%) | 1.5 (1.1–1.9)[‡] | 1.4 (1.1–1.9)[‡] |
| 4th Quartile (Richest) | 102 (26%) | 2,769 (24%) | 1.4 (1.0–1.8) | 1.4 (1.0–1.8) |
| **Highest Education Level (age 15+)** | | | | |
| None | 102 (27%) | 2,039 (18%) | Baseline | Baseline |
| Primary | 195 (51%) | 6,056 (52%) | 0.6 (0.5–0.8) | 0.9 (0.7–1.1) |
| Secondary | 80 (21%) | 3,128 (27%) | 0.5 (0.4–0.7)[‡] | 0.7 (0.5–0.9)[‡] |
| University | 3 (1%) | 348 (3%) | 0.2 (0.5–0.5)[‡] | 0.2 (0.5–0.9)[‡] |
| **Literacy (age 15+)** | | | | |
| Can Read Well | 188 (49%) | 5,986 (52%) | 1.0 (0.8–1.3) | 0.9 (0.7–1.2) |
| Can Read a Little | 114 (30%) | 2,956 (25%) | 1.2 (1.0–1.6) | 1.2 (1.0–1.6) |
| Cannot Read at all | 83 (22%) | 2,677 (23%) | Baseline | Baseline |
| **Marital Status (15+)** | | | | |
| Married/living together | 175 (56%) | 4,922 (59%) | Baseline | Baseline |
| Divorced/separated | 27 (9%) | 319 (4%) | 2.4 (1.6–3.6)[‡] | 2.0 (1.3–3.0)[‡] |
| Widowed | 34 (11%) | 395 (5%) | 2.4 (1.7–3.5)[‡] | 1.6 (1.0–2.4)[‡] |
| Never married/lived with another | 76 (24%) | 2,696 (32%) | 0.8 (0.6–1.0) | 0.8 (0.6–1.2) |
| **Other functional limitations** | | | | |
| Vision | 60 (16%) | 201 (2%) | 10.5 (7,7–14.3)[‡‡] | 8.5 (6.1–11.8)[‡‡] |
| Hearing | 33 (9%) | 100 (1%) | 10.8 (7.2–16.2)[‡‡] | 8.5 (5.6–13.1)[‡‡] |
| Walking | 83 (22%) | 210 (2%) | 14.9 (11.3–19.7)[‡‡] | 13.3 (9.7–18.3)[‡‡] |
| Self-Care | 35 (9%) | 69 (1%) | 16.7 (10.9–25.4)[‡‡] | 13.2 (8.5–20.5)[‡‡] |

*(Continued)*

**Table 3.** (Continued)

| | People with anxiety and/or depression (n = 385) | People without anxiety and/or depression (n = 12,688) | Unadjusted OR (95% CI) | Age, sex, adjusted OR (95% CI) |
|---|---|---|---|---|
| | N (%) | N (%) | | |
| Communication or cognition | 81 (21%) | 147 (1%) | 20.8 (15.5–27.9) [‡‡] | 17.6 (13.0–23.8) [‡‡] |

[‡‡]p<0.001
[a]0.05

mental health disorders on other health conditions that has been show in previous studies [20]. Cases with anxiety and/or depression had four times the odds of receiving medication for 'Depression or anxiety' within the past 12 months compared to cases without (aOR 4.1, 1.9–9.1). They also had nearly twice the odds of receiving medication for 'Sleep problems' within the past 12 months compared to cases without (aOR 1.9, 1.1–3.5). Of 385 cases with anxiety and/or depression, 25 (6%) had received medication for depression or anxiety in the past 12 months.

## Healthcare access

Table 6 compares health advice and treatment between cases with and without anxiety and/or depression. Again, a range of indicators were included due to the broad impact of mental health conditions on healthcare access shown in past studies [27]. Cases with anxiety and/or depression were nearly twice as likely to report a serious health problem in the last 12 months

**Table 4.** Education (5–17) and livelihood (18+), comparing cases with and without anxiety and/or depression.

| | Child case with anxiety and/or depression (n = 85) | Child case without anxiety and/or depression (n = 46) | | |
|---|---|---|---|---|
| | N (%) | N (%) | Unadjusted OR (95% CI) | Age, sex, region, SES adjusted OR (95% CI) |
| Education (5–17) | | | | |
| Currently enrolled in school | 66 (78%) | 32 (70%) | 1.5 (0.7–3.4) | 1.5 (0.6–3.7) |
| Not currently enrolled | 19 (23%) | 14 (30%) | Baseline | Baseline |
| | Adult case with anxiety and/or depression (n = 229) | Adult case without anxiety and/or depression (n = 349) | | |
| Livelihood (18+) | | | | |
| Worked in past 7 days | 86 (38%) | 108 (31%) | 1.3 (0.9–1.9) | 1.3 (0.9–1.9) |
| Not worked in past 7 days | 143 (62%) | 241 (69%) | Baseline | Baseline |
| | All cases with anxiety and/or depression (n = 385) | All cases without anxiety and/or depression (n = 548) | | |
| State Benefits | | | | |
| Retirement pension | 9 (4%) | 40 (12%) | 0.3 (0.2–0.6)[‡] | 0.4 (0.2–0.8)[‡] |
| Disability Pension | 2 (1%) | 6 (2%) | 0.5 (0.1–2.5) | 0.5 (0.1–2.6) |
| Family Allowance | 35 (15%) | 46 (13%) | 1.2 (0.7–1.9) | 1.5 (0.9–2.4) |
| Other | 7 (3%) | 9 (3%) | 1.2 (0.4–3.2) | 1.4 (0.5–3.9) |

[‡‡]p<0.001
[‡]p<0.05

**Table 5. Medication/treatment for health conditions diagnosed by doctor, comparing cases with and without anxiety and/or depression.**

| | Received medication for condition in the past 12 months. | | | |
|---|---|---|---|---|
| | Case with anxiety and/or depression (n = 385) | Case without anxiety and/or depression (n = 548) | | |
| | N (%) | N (%) | Unadjusted OR (95% CI) | Age, sex, region, SES adjusted OR (95% CI) |
| Vision loss. | 18 (5%) | 31 (6%) | 0.8 (0.5–1.5) | 1.0 (0.6–1.9) |
| Hearing loss. | 2 (1%) | 11 (2%) | 0.3 (0.1–1.2) | 0.3 (0.1–1.3) |
| Arthritis, arthrosis. | 20 (5%) | 25 (5%) | 1.1 (0.6–2.1) | 1.6 (0.8–2.9) |
| Heart disease, coronary disease, heart attack. | 20 (5%) | 22 (4%) | 1.3 (0.7–2.4) | 1.6 (0.8–3.0) |
| Chronic bronchitis or emphysema. | 18 (5%) | 17 (3%) | 1.5 (0.8–3.0) | 1.7 (0.9–3.5) |
| Asthma, allergic respiratory disease. | 14 (4%) | 17 (3%) | 1.2 (0.6–2.4) | 1.2 (0.6–2.6) |
| Back pain or disc problems. | 22 (6%) | 24 (4%) | 1.3 (0.7–2.4) | 1.4 (0.8–2.7) |
| Migraine (recurrent headaches). | 44 (11%) | 50 (9%) | 1.2 (0.7–1.8) | 1.2 (0.7–1.8) |
| Stroke (i.e. cerebral bleeding). | 6 (2%) | 17 (3%) | 0.5 (0.2–1.3) | 0.7 (0.3–1.7) |
| Depression or anxiety. | 25 (6%) | 9 (2%) | 4.2 (1.9–9.0) [‡‡] | 4.1 (1.9–9.1) [‡‡] |
| Tumour or cancer (including blood cancer). | 4 (1%) | 2 (1%) | 2.8 (0.5–15.7) | 4.1 (0.7–24.2) |
| Dementia. | 0 | 0 | - | - |
| Kidney diseases. | 14 (4%) | 10 (2%) | 2.0 (0.9–4.6) | 2.2 (0.9–5.1) |
| Skin diseases e.g. psoriasis. | 14 (4%) | 13 (2%) | 1.6 (0.7–3.3) | 1.7 (0.8–3.7) |
| Tuberculosis. | 2 (1%) | 0 | - | - |
| Mental (psychiatric) or behavioural disorders. | 0 | 2 (1%) | - | - |
| Sleep problems. | 25 (6%) | 24 (4%) | 1.5 (0.9–2.7) | 1.9 (1.1–3.5) [‡] |
| Tinnitus | 5 (1%) | 9 (2%) | 0.8 (0.3–2.4) | 0.8 (0.3–2.4) |
| Severe diarrhea | 18 (5%) | 21 (4%) | 1.2 (0.6–2.3) | 1.3 (0.7–2.6) |
| Perinatal complications | 3 (1%) | 1 (0%) | 4.2 (0.4–41.5) | 3.3 (0.3–32.2) |
| Malnutrition | 5 (1%) | 6 (1%) | 1.2 (0.4–3.9) | 1.2 (0.3–4.0) |
| Mosquito borne illness (dengue, malaria, chikungunya, zika) | 34 (9%) | 52 (9%) | 0.9 (0.6–1.5) | 0.8 (0.5–1.3) |
| Has received medication for condition. | | | | |
| Diabetes | 19 (5%) | 29 (5%) | 0.9 (0.5–1.7) | 1.2 (0.7–2.3) |
| Hypertension | 49 (13%) | 67 (12%) | 1.0 (0.7–1.6) | 1.4 (0.9–2.1) |

[‡‡]p<0.001

[‡]p<0.05

compared to cases without anxiety and/or depression (aOR 1.8, 1.3–2.5). Of 385 cases with anxiety and/or depression, 161 (51%) had a serious health problem in the last 12 months. Cases with anxiety and/or depression were much more likely than cases without anxiety and/or depression to have sought advice/treatment for a health problem with a Government Community Health Worker/Health Post (aOR 6.3, 1.0–39.2), a Private Clinic/Hospital (aOR 3.1, 1.3–7.4), or another place (aOR 3.5, 1.0–11.6). Feeling respected, understanding information, and being understood were not significantly associated with anxiety and/or depression.

## Discussion

This study had the objectives of determining the sociodemographic characteristics of people with anxiety and/or depression in Guatemala, and what, if any, increased burden they face

**Table 6. Health advice/treatment, comparing cases with and without anxiety and/or depression.**

| | Case with anxiety and/or depression (n = 385) | Case without anxiety and/or depression (n = 548) | Unadjusted OR (95% CI) | Age, sex, region, SES adjusted OR (95% CI) |
|---|---|---|---|---|
| | N (%) | N (%) | | |
| **Serious health problem(s) in the last 12 months.** | | | | |
| Yes | 161 (51%) | 172 (44%) | 1.4 (1.0–1.8) [‡‡] | 1.8 (1.3–2.5) [‡‡] |
| **Sought advice or treatment for problem(s).** | | | | |
| Yes | 122 (76%) | 132 (77%) | 0.9 (0.6–1.6) | 0.8 (0.4–1.3) |
| **Last time sought advice/treatment for health problem(s)*** | | | | |
| *Where sought* | | | | |
| Govt Health Centre | 19 (16%) | 28 (21%) | | Baseline |
| Govt Community Health Worker/ Health Post | 8 (7%) | 2 (2%) | 5.9 (1.1–30.8) [‡] | 6.3 (1.0–39.2) [‡] |
| Govt/IGSS Hospital | 31 (34%) | 52 (39%) | 1.1 (0.6–2.4) | 1.5 (0.7–3.4) |
| Pharmacy | 5 (4%) | 19 (14%) | 0.4 (0.1–1.2) | 0.5 (0.1–1.5) |
| Private Clinic/Hospital | 38 (31%) | 24 (18%) | 2.3 (1.1–5.1) [‡] | 3.1 (1.3–7.4) [‡] |
| Other | 11 (9%) | 7 (5%) | 2.3 (0.7–7.0) | 3.5 (1.0–11.6) [‡] |
| *Feeling respected* | | | | |
| Completely or mostly respected | 203 (64%) | 232 (59%) | Baseline | Baseline |
| Neither respected nor disrespected | 19 (6%) | 37 (9%) | 0.6 (0.3–1.1) | 0.6 (0.3–1.2) |
| Completely or mostly disrespected | 16 (5%) | 31 (8%) | 0.6 (0.3–1.1) | 0.7 (0.3–1.3) |
| Does not apply (has never sought advice) | 76 (24%) | 93 (24%) | 0.9 (0.7–1.3) | 0.8 (0.6–1.2) |
| *Understanding information* | | | | |
| Easy | 143 (46%) | 165 (42%) | Baseline | Baseline |
| Neither easy nor difficult | 41 (13%) | 68 (17%) | 0.7 (0.4–1.1) | 0.8 (0.5–1.2) |
| Difficult | 56 (18%) | 65 (17%) | 1.0 (0.6–1.5) | 1.1 (0.7–1.7) |
| Does not apply (has never sought advice) | 76 (24%) | 93 (24%) | 0.9 (0.6–1.1) | 0.8 (0.5–1.2) |
| *Being understood* | | | | |
| Easy | 140 (45%) | 161 (41%) | Baseline | Baseline |
| Neither easy nor difficult | 41 (13%) | 68 (17%) | 0.7 (0.4–1.1) | 0.8 (0.5–1.2) |
| Difficult | 54 (17%) | 65 (17%) | 1.0 (0.6–1.5) | 1.1 (0.7–1.7) |
| Does not apply (has never sought advice) | 76 (24%) | 93 (24%) | 0.9 (0.6–1.1) | 0.8 (0.6–1.3) |

*Amongst those who report having sought advice

[‡‡]p<0.001

[‡]p<0.05

relative to other people with disabilities. Age, sex, region, location, education, marital status, and other functional limitations were all significantly associated with anxiety and/or depression. Compared to people with disabilities, not including anxiety/depression, those with anxiety/depression were significantly less likely to receive a retirement pension. They were also significantly more likely to receive medication for depression/anxiety or sleep problems, report a serious health problem, and seek advice/treatment with a government health worker/health post, private clinic/hospital, or another place.

Epidemiological data on mental health disorders in Guatemala are needed to inform the implementation of the country's National Mental Health Plan. ENDIS was Guatemala's first national, population-level study to measure impaired functioning using the WG-ES

instrument. It presented an opportunity to better understand the sociodemographic characteristics and service access of people with anxiety and/or depression. ENDIS was not highly comparable with the 2009 ENSM due to the difference in data collection instruments. It was also conducted seven years after ENSM, and had a substantially larger sample size. In spite of these differences, the ENSM study is the most scientifically-rigorous point of comparison to ENDIS [4]. Another less-direct comparison is the 2001–2002 National Survey on Psychiatric Epidemiology (ENEP) conducted in Mexico, a country sharing a border and key sociodemographic characteristics with Guatemala. ENEP was conducted 15 years before ENDIS, and like ENSM, used the CIDI instrument [28]. Unfortunately other population-level data on mental health disorders in Guatemala are scarce, making comparisons over time difficult. The recent 2018 National Census in Guatemala regrettably did not include the WG questions on anxiety and/or depression [10]. This highlights the need for continued, population-level research on mental health in Guatemala.

In this study we found that people aged 50+ had over twice the likelihood of anxiety and/or depression as those aged 5–17. By comparison, the 2009 ENSM study in Guatemala found the highest prevalence of both affective and anxiety disorders to be in the 21 to 50 age group [4]. Furthermore, the 2019 Global Burden of Disease Study found that, globally, disability-adjusted life years (DALYs) attributed to depressive and anxiety disorders peaked in the 30–34 year age range [29]. This difference may be due to the methodological differences between the CIDI and WG instruments, however it could also be due to the ageing of the internal conflict-affected population in Guatemala. According to the 2018 National Census, 14.7% of the population falls into the 50+ age range [10]. There is an increased need for mental health services targeting the 50+ segment of the population. We also found that females had nearly twice the odds of anxiety and/or depression as males, and that people living in an urban area had one and a half times the odds as those living in a rural area. Both of these findings are consistent with those of the ENSM study [4]. The ENEP study also found that females were more likely to have affective and anxiety disorders, and that residents of metropolitan areas were more likely to have anxiety disorders, but not affective disorders, in the past month [28]. These findings highlight the need for mental health services for women and residents of urban areas.

Living in the Southeast and Northeast regions of the country was also associated with significantly decreased odds of anxiety and/or depression compared to living in the Central region. Mental health-focused policies and services should prioritize the Central and Northwest regions of the country.

Functional difficulties with vision, hearing, mobility, cognition or communication was associated with significantly increased odds of anxiety/depression. Similar findings of strong association between anxiety/depression and difficulties with physical and sensory functioning have been demonstrated in LMICs. Wallace et al. hypothesized that reasons for this could include '. . . .pain, reduced perceived control, activity restriction, impact on financial circumstances and changing social relationships. . .' These associations highlight the need for services that can address physical and mental health difficulties together. Further research is needed to determine whether there is a causal relationship between affect and non-affect impairments [25].

Our study found that people with a university level of education were significantly less likely to have anxiety and/or depression than people with no formal education; it also found that literacy was not significantly associated with anxiety and/or depression. However, ENSM found that people who "Know how to read and write" had a higher likelihood of affective and anxiety disorders [4]. This disagreement calls for further research on the relationship between low literacy and anxiety and/or depression, considering a review by Maughan et al. which found some co-occurrence of reading failure with anxiety and depression [30]. We also found that

people who are divorced/separated or widowed had higher odds of anxiety and/or depression when compared to those who were married/living together. This finding is consistent with other studies showing links between family relationships and mental health and specifically a positive association between marriage and mental health [31, 32].

We found no significant association between ethnic group and anxiety and/or depression in this study. This was unexpected given that ENSM found greatly-increased odds of having depression or post-traumatic stress disorder (PTSD) in indigenous people who had been exposed to violence when compared to non-indigenous people who had also been exposed to violence [33]. Another study, however, found that "socioeconomic and health-related variables" greatly confounded the relationship between ethnic group and anxiety and/or depression [31, 34]. We found no association between SES and anxiety and/or depression in this study. This was surprising given the expanding evidence on the inverse relationship between SES and depression [35]. This may relate to the SES measure. We used an asset based measure, in contrast to many other studies that have used income, occupation, education level, social class or as SES indicators [36]. It is possible that this asset based SES measure was too blunt a tool to measure poverty in these settings. Further, asset-based measures are more reflective of long-term economic well-being, and so may not reflect changes in wealth due to recent onset of disability. Our current analysis did find an association with education level which is another indicator of socio-economic status [36]. Further research using different socioeconomic status indicators is needed in this setting to clarify these findings.

Six percent of people with anxiety and/or depression in Guatemala received medication for depression or anxiety in the past 12 months. This extremely low rate of medication likely reflects low access to mental health services in the Guatemalan population. Similarly, the ENSM study found that very few (just over 2%) people with mental health issues access "any [mental health] service" [4]. The ENEP study in Mexico showed a better rate of access to "any service" for people with affective and anxiety disorders, at 20% and 12%, respectively [28]. Once again, these numbers are not completely comparable due to different data collection methods, and furthermore ENDIS did not measure other mental health and psychosocial services (MHPSS). However, they are consistent with evidence of a major treatment gap in mental health in the Region of the Americas [7]. Our findings highlight a likely treatment gap for people with anxiety and/or depression in Guatemala. Further research is needed regarding access to other MHPSS, like inpatient, outpatient, and community-based mental health services.

Over half of those with anxiety and/or depression had a serious health problem in the past 12 months, and this was significantly higher than people with non-affect impairments. This indicates that having anxiety and/or depression is associated with greater risk of health problems. Existing epidemiological studies of mental health concur with this result, widely showing that having a mental health issue increases one's risk of having a physical health condition, and vice versa [27]. Compared with people without anxiety and/or depression, people with anxiety and/or depression were over six times as likely to seek healthcare support from a community health post than a government health centre. There is increasing evidence on the value of community mental health services and task-sharing, whereby primary health care workers are trained to deliver mental health care. Our finding, of higher utilisation of community health posts therefore suggests that training and equipping community health workers staffing these posts to provide mental health services in addition to physical health services may be an important intervention to address their co-occurrence and the mental health treatment gap [37, 38].

This study had some key limitations that are important to note. Firstly, there are limitations relating to the assessment of anxiety and depression. We used self-report questions from the WG-ES and CFM rather than either clinical assessment or using validated clinical screening

tools (e.g. K6, PHQ-9 or GAD-9) [25, 39, 40]. There is some evidence of an association between anxiety and depression identified using the WG questions and mental health screening tools (K6 and GAD) and we found an association between having anxiety/depression and being female and lower education levels; this trend aligns with many previous studies, lending some weight to the findings [26]. However, research into the validity and reliability of these WG questions on anxiety/depression is limited and findings from other studies have suggested that they may under-estimate symptomatic anxiety and depression [41]. The WG/CFM questions ask directly about feelings of anxiety and depression, which differs to other anxiety/depression screening tools which more commonly ask about severity of different symptoms of these conditions. Although the questions underwent forward/backward translations and pilot testing in the different languages, we cannot rule out different cultural interpretations of the terms 'anxiety' or 'depression' (in the questions), or that stigma related to these conditions may also have also resulted in response bias and likely under-reporting. There is a need for more rigorous research into the validity and reliability of these questions in different settings. Another limitation is that this was a cross sectional survey and data were not collected on age of onset of anxiety/depression and it is not possible to establish the causality or temporality of the associations with exposure variables (e.g. marital status).

This study only assessed anxiety and depression and not other categories of mental health disorders like CIDI did. The data thus did not indicate the relative importance of depression/anxiety compared to other mental health disorders.

Despite these limitations, our findings that women, older people, people with lower educational levels and those with functional difficulties are at increased risk of anxiety and depression align with previous studies and suggest that access to mental health and psychosocial support services for these groups deserves particular attention. Although this study did not explore access to mental health services in depth, our findings suggest that only 6% of people with anxiety/depression had received related services/treatment. This aligns with other studies which show substantial gaps in the treatment of mental health disorders, caused by barriers impeding providers from delivering their services, and users from accessing them [42]. People with functional limitations in other domains may face additional barriers to accessing services (e.g. physical inaccessibility and communication challenges) [25, 43]. Drawing on increasing evidence on effectiveness of community-based approaches and clinical task sharing, and considering the trend in this study that people with anxiety/depression were more likely to utilise community health posts, it is recommended that attention be paid to strengthening and developing community level mental health and psychosocial support services in this setting [44].

In conclusion, the public health system of Guatemala should take action to strengthen access to mental health services with particular attention to older adults, women, residents of urban areas, those with lower education levels, and people with other types of functional limitations.

## Acknowledgments

We thank the following individuals/institutions for their contribution to this project:

Participants, for participating in data collection.

Field staff, for conducting data collection.

CONADI, particularly Carlos Dionicio, for managing data collection, analyzing, and writing-up the study.

LSHTM, particularly Islay Mactaggart, Sarah Polack, and Jonathan Naber, for designing, supporting, analyzing, and writing-up the study.

## Author Contributions

**Conceptualization:** Jonathan Naber, Islay Mactaggart, Carlos Dionicio, Sarah Polack.

**Data curation:** Islay Mactaggart, Sarah Polack.

**Formal analysis:** Islay Mactaggart, Sarah Polack.

**Funding acquisition:** Islay Mactaggart, Sarah Polack.

**Methodology:** Jonathan Naber, Islay Mactaggart, Carlos Dionicio, Sarah Polack.

**Project administration:** Jonathan Naber, Islay Mactaggart, Carlos Dionicio.

**Supervision:** Jonathan Naber, Carlos Dionicio.

**Writing – original draft:** Jonathan Naber, Islay Mactaggart, Carlos Dionicio.

**Writing – review & editing:** Jonathan Naber, Islay Mactaggart, Carlos Dionicio, Sarah Polack.

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
