## [Decision Letter · Decision Letter 0]

1 Oct 2021

PONE-D-21-20741Anxiety and depression in Guatemala: sociodemographic characteristics and service accessPLOS ONE

Dear Dr. Naber,

Thank you for submitting your manuscript to PLOS ONE. After careful consideration, we feel that it has merit but does not fully meet PLOS ONE’s publication criteria as it currently stands. Therefore, we invite you to submit a revised version of the manuscript that addresses the points raised during the review process.

ACADEMIC EDITOR: This is an interesting paper and is potentially a very useful addition to the literature. But need to address some critical points as reviewers pointed out.==============================

We look forward to receiving your revised manuscript.

Kind regards,

Seo Ah Hong, PhD

Academic Editor

PLOS ONE

“CONADI, CBM, and UNICEF for financing this study.”

“The Guatemala survey was funded by CBM Latin America and CONADI (the National Council on Disability), in a grant to Dr. Sarah Polack. No specific funding was received for these analyses of the data.”

5.We note that you have indicated that data from this study are available upon request. PLOS only allows data to be available upon request if there are legal or ethical restrictions on sharing data publicly. For more information on unacceptable data access restrictions, please see http://journals.plos.org/plosone/s/data-availability#loc-unacceptable-data-access-restrictions.

7. Your ethics statement should only appear in the Methods section of your manuscript. If your ethics statement is written in any section besides the Methods, please delete it from any other section.

Additional Editor Comments (if provided):

Reviewers' comments:

Reviewer's Responses to Questions

**Comments to the Author**

1. Is the manuscript technically sound, and do the data support the conclusions?

Reviewer #1: Yes

Reviewer #2: No

2. Has the statistical analysis been performed appropriately and rigorously? 

Reviewer #1: Yes

Reviewer #2: No

3. Have the authors made all data underlying the findings in their manuscript fully available?

Reviewer #1: Yes

Reviewer #2: No

4. Is the manuscript presented in an intelligible fashion and written in standard English?

Reviewer #1: Yes

Reviewer #2: Yes

5. Review Comments to the Author

Reviewer #1: General comments:

This article contribute to the lack of epidemiological data on mental health disorders in Guatemala. Studies like this are crucial to improve people’s life through the needed policy changes at the regional and national level.

It is important to notice that the data collection took place in 2016, reducing is validity and direct implications. However, there is a high need of mental health data from Guatemala and other neighboring countries that make the study significant.

Another major concern is the assessment selected. Even though it is promising, The Washington Group extended set on functioning (WG-ES) is s recent tool. I recommend a more extended description of the tool and justification of its use. Also to include any available data on psychometric characteristics of the survey along with validity and comparison with other more commonly used assessments of mental health disorders.

The discussion does not go deep on the inconsistent findings. More reflection and potential hypothesis should be presented to better discuss the interesting results.

Specific Comments:

Introduction:

Page 4: While this study generated important data for Guatemala, it only included adults between the ages of 18 and 65, effectively excluding a large part of the population.4

Population from 18-65 does not exclude a large part of the general population. Please be more accurate with the statements. Maybe you can say excluding older adults which represent and children X % of the population.

Methods:

Page 6: Self-reported functional difficulties

This is a very vague term. Is this the formal term used by the questionnaire? If so, please explain. Starting the case definition with this term reduces the article credibility.

Page 7: Variables description.

More information about the variables is needed. From the main outcomes to the covariates. What combined depression and anxiety means? Explain outcome variables and scoring in detail to increase test validity (and author’s credibility). It would be useful to have more information about the survey questions. For example, number of items, examples of questions related to access to health services or livelihood, ect. More information about the variables is needed.

Page 8: The WG instrument was translated into the four leading non-Spanish languages of Guatemala - K’iche’, Kaqchikel, Mam, and Q’eqchi’.

Was this translation process similar to the English to Spanish or was it translated directly from the Spanish translation? Did you calculate internal validity of the tool in the different languages? Any information about its psychometric characteristics would be useful.

Page 8: Two weeks of mop-up were conducted at the end of the survey in clusters with low response rates or incomplete data, primarily in the Central and Northwest regions.

In what this consisted. I guess that if you mention it is because it is enough important to be better explained. Please provide further details.

Results:

The age distribution is very limited. The sample is not evenly distributed at all after all the sampling described in the methods. From the 13073 total sample, only 2035 are above 50 years old. Is it representative of the Guatemaltecan population (15% of people are >50).

Page 14: However, of 385 cases with impaired affect, only 25 (6%) had received medication for depression or anxiety in the past 12 months.

My recommendation is to avoid any judgement in the results section and leave them for the discussion.

Discussion:

Page 19: There is an increased need for mental health services targeting the 50+ segment of the population.

What percentage of the total population it represents? What kind of disorders they may be facing. What are the implications or future studies related ot that?

In general, results could be compared with other similar studies/surveys from neighboring countries or at least other Latino countries.

Page 20: it also found that literacy was not significantly associated with impaired affect. However, ENSM found that people who ‘’Know how to read and write’’ had a higher likelihood of affective and anxiety disorders.4 This disagreement calls for further research on the relationship between low literacy and impaired affect, considering a review by Maughan et al. which found some co-occurrence of reading failure with anxiety and depression.24

It would be interesting to know, if available, how many people had a low literacy level from the survey and how the informed consent was done on those cases. If you have this data, it would be useful in order to increase the information about the results.

Page 20: We found no significant association between ethnic group and impaired affect in this study. This was unexpected given that ENSM found greatly-increased odds of having depression or post-traumatic stress disorder (PTSD) in indigenous people who had been exposed to violence when compared to non-indigeous people who had also been exposed to violence.27 Another study, however, found that “socioeconomic and health-related variables” greatly confounded the relationship between ethnic group and impaired affect.25,28 We found no association between SES and impaired affect in this study. This was surprising given the expanding evidence on the inverse relationship between SES and depression.29 Further research on ethnicity and socioeconomic status is needed to clarify these unexpected results.

These are indeed very odd results. Please expand on potential explanations and what the literature from other countries says on this respect as it questions the validity of the study (test validity). For example, is there any literature about resilient factors among indigenous population that may explain that 9 years later this subgroup is doing better? At least, you should mention the limitation of the lack of use of this survey and culturally adapted tools for these subgroups as this might be the reason for not finding any differences.

Page 21: Community health workers staffing these posts should be trained and equipped to provide mental health services in addition to physical health services to address their co-occurrence.

If the authors decide to recommend interventions, more justifications and detail should be provided. This sentence should at least be referenced. See “Alegría M, et al. Effectiveness of a Disability Preventive Intervention for Minority and Immigrant Elders: The Positive Minds-Strong Bodies Randomized Clinical Trial. Am J Geriatr Psychiatry. 2019 Dec;27(12):1299-1313. doi: 10.1016/j.jagp.2019.08.008. Epub 2019 Aug 13. PMID: 31494015; PMCID: PMC6842701.” For a useful reference.

Page 21: or from ‘.... cultural differences in interpretation of the tools….’

I don’t understand what this sentence means in the middle of the discussion. Is this a qualitative quote from an investigator? This sentence lacks scientific rigor. Please review and correct for accuracy and add references to discuss this important aspect of the study. This is the most important limitation.

Page 22: Conclusions

Some indication to the policy level needed changes should be included here. How these results can serve the policy makers to improve the quality of life and services among people with impaired affect?

Reviewer #2: General comments

The study deals with a relevant topic and is based on national epidemiological data. However, there are several issues that are unclear and make it difficult to understand the theoretical framework, methods, results and conclusions.

The term “impaired affect” is inadequate to designate people with anxiety and depressive symptoms. This issue is central to the study, as the authors assume this concept, including the terms anxiety and depression in the title, when in fact there is no formal assessment of these symptoms/disorders.

Specific comments

Introduction:

The correct name of the instrument used (CIDI) is not Compact International Diagnostic Interview, but Composite International Diagnostic Interview.

The entire theoretical framework of the study is based on the lack of studies/epidemiological data on mental health in Guatemala. The authors review these studies and describe the prevalence of mental disorders, showing that anxiety/somatoform and mood disorders are the most prevalent. However, there is no description of studies that have evaluated impaired affect and we cannot infer that these can be automatically translated into anxiety and mood disorders.

Thus, there is an important gap in the theoretical framework, which does not allow us to support the assertion that “The substantial gap in mental health spending and treatment in Guatemala suggests that people with impaired affect may face heightened burdens compared to people with other disabilities or with no disabilities.”

Methods

Case definition

The instrument/questionnaire used to define the case should be described in greater detail. How was this questionnaire constructed? It would be important to include references to the instrument.

The authors describe that both anxiety and depression were assessed using two questions, which included frequency (daily and a lot) for their classification. It would be important for the authors to provide references for the criteria adopted for the definition of impaired affect considering these questions. Which studies have used these criteria?

Returning to the question raised above, the definition of impaired affect, based on direct questions about feelings of anxiety, nervousness, worry and depression, and the frequency of such feelings, seems inadequate to me.

The study does not describe how exposure variables were measured.

Another methodological problem lies in the fact that there is no time definition for the onset of symptoms that led to the case definition. So how do we know about the exposure window for cases and controls? It would be correct to make a pairing of cases and controls, so that the onset date of the symptoms that classify the cases was taken to the time of exposure to the exposure variables also for the controls. In the current, nested case-control format, we have no way of knowing the temporality between a series of evaluated exposures and the outcome under study, such as marital status (for example, which came first, separation/divorce or mental disorder?), or even other variables. Such a format can lead to important biases.

Other important points:

The description of the study design (nested case-control) is insufficient. Include more details on eligibility criteria, etc. Were all non-cases included in the study?

The inclusion of children – how was data collection conducted? There is no indication whether there has been an adaptation of the instrument, or whether it is suitable for children aged 5 years and over. Has any kind of assessment been made of the validity of questions about anxiety and depression symptoms for such young children? There is a straightforward question about depression. How does a child interpret this?

My suggestion is that the study only includes people aged 18 and over.

Results

The format for presenting the results is confusing. “Impaired affect” is the outcome under study; therefore, the tables should present the prevalence of the outcome according to the categories of the exposure variables. What the tables show is the frequency of exposures according to the outcomes, which is not adequate to meet the proposed objectives.

The results should include the crude and adjusted odds ratios.

Discussion

The study has limitations that have been insufficiently discussed, namely:

Diagnostic criteria for defining anxiety/depression:

The fact that WG-ES is based on self-report is an important limitation, but not the main one. In the case of the present study, not determining the onset of symptoms leads to problems that can lead to biases in the estimates found:

Temporality – there is no guarantee that some exposures occurred before the outcome.

Exposure windows for cases and controls may be different. If there was information about the date of onset of symptoms, there would be a possibility of pairing cases and controls, considering the exposure time in the paired control to be the same as in the case.

The comparison with the results found in other studies does not consider differences in these criteria, which can lead to differences in the prevalence of the outcomes under study.

6. PLOS authors have the option to publish the peer review history of their article (what does this mean?). If published, this will include your full peer review and any attached files.

Reviewer #1: No

Reviewer #2: No

---

## [Author Response · Author response to Decision Letter 0]

18 Mar 2022

Please see attached 'Response to Reviewers.'

---

## [Decision Letter · Decision Letter 1]

14 Jun 2022

PONE-D-21-20741R1Anxiety and depression in Guatemala: sociodemographic characteristics and service accessPLOS ONE

Dear Dr. Naber,

Thank you for submitting your manuscript to PLOS ONE. After careful consideration, we feel that it has merit but does not fully meet PLOS ONE’s publication criteria as it currently stands. Therefore, we invite you to submit a revised version of the manuscript that addresses the points raised during the review process.

Thanks for the revision.

A reviewer requests to revise a few minor issues so please address them. And a few minor things should be corrected as follows.

1. Revise from “Cases (indicating ‘A lot’ or ‘Cannot do’ in 1+ functional domain) and age-/sex-matched controls were administered a structured questionnaire” to “Cases (indicating ‘A lot of difficulty’ or ‘Cannot do’ in one or more functional domain) and age-/sex-matched controls were administered a structured questionnaire. “ in the Abstract. (Page 2, Line 16)

2. Delete “(WG-ES)” and “(CFM)” in the sentences “The WG ES was developed, tested and adopted by the Washington Group on Disability Statistics (WG-ES)” and “The CFM was developed and tested by the WG together with UNICEF (CFM).” (Page 6, Line 101-2)

3. “We found no significant association between ethnic group and impaired affect in this study.” And additional 5-6 places having the term in the text.

=>correct “impaired affect” into “anxiety and/or depression”

4. Correct from “with/out” to “without” (Page 10, Line 176)

We look forward to receiving your revised manuscript.

Kind regards,

Seo Ah Hong, PhD

Academic Editor

PLOS ONE

Journal Requirements:

Reviewers' comments:

Reviewer's Responses to Questions

**Comments to the Author**

1. If the authors have adequately addressed your comments raised in a previous round of review and you feel that this manuscript is now acceptable for publication, you may indicate that here to bypass the “Comments to the Author” section, enter your conflict of interest statement in the “Confidential to Editor” section, and submit your "Accept" recommendation.

Reviewer #1: All comments have been addressed

Reviewer #3: All comments have been addressed

2. Is the manuscript technically sound, and do the data support the conclusions?

Reviewer #1: Yes

Reviewer #3: Yes

3. Has the statistical analysis been performed appropriately and rigorously? 

Reviewer #1: Yes

Reviewer #3: Yes

4. Have the authors made all data underlying the findings in their manuscript fully available?

Reviewer #1: Yes

Reviewer #3: (No Response)

5. Is the manuscript presented in an intelligible fashion and written in standard English?

Reviewer #1: Yes

Reviewer #3: Yes

6. Review Comments to the Author

Reviewer #1: The authors have responded to all my specific comments. The article has improved after the revision and the topic is significant enough to be published at PLOS ONE

Reviewer #3: Excellently written article. Case control design and analysis are well designed and systematically carried out.

Tables are very clear.

Discussion refers to the references and findings.

Minor point: What is the reason to include 5-17 group? (Table3) The sample description in the table 2 showed 0-14 groups.

If authors are serious to add these age group, I think the logic to include them should be discussed.

Otherwise comparing children and old age may not make sense.

7. PLOS authors have the option to publish the peer review history of their article (what does this mean?). If published, this will include your full peer review and any attached files.

Reviewer #1: No

Reviewer #3: **Yes: **Myo Nyein Aung

---

## [Author Response · Author response to Decision Letter 1]

21 Jun 2022

Please see attached 'Response to Reviewers _ 20 June 2022.'

---

## [Decision Letter · Decision Letter 2]

27 Jul 2022

Anxiety and depression in Guatemala: sociodemographic characteristics and service access

PONE-D-21-20741R2

Dear Dr. Naber,

We’re pleased to inform you that your manuscript has been judged scientifically suitable for publication and will be formally accepted for publication once it meets all outstanding technical requirements.

Kind regards,

Seo Ah Hong, PhD

Academic Editor

PLOS ONE

Additional Editor Comments (optional):

Reviewers' comments:

Reviewer's Responses to Questions

**Comments to the Author**

1. If the authors have adequately addressed your comments raised in a previous round of review and you feel that this manuscript is now acceptable for publication, you may indicate that here to bypass the “Comments to the Author” section, enter your conflict of interest statement in the “Confidential to Editor” section, and submit your "Accept" recommendation.

Reviewer #3: All comments have been addressed

2. Is the manuscript technically sound, and do the data support the conclusions?

Reviewer #3: Yes

3. Has the statistical analysis been performed appropriately and rigorously? 

Reviewer #3: Yes

4. Have the authors made all data underlying the findings in their manuscript fully available?

Reviewer #3: Yes

5. Is the manuscript presented in an intelligible fashion and written in standard English?

Reviewer #3: Yes

6. Review Comments to the Author

Reviewer #3: I found that the authors have addressed the reviewers' comment and revised the article as much as they can improve.

It is acceptable in current situation to be published.

7. PLOS authors have the option to publish the peer review history of their article (what does this mean?). If published, this will include your full peer review and any attached files.

Reviewer #3: **Yes: **Myo Nyein Aung

---

## [Editor Report · Acceptance letter]

5 Aug 2022

PONE-D-21-20741R2 

Anxiety and depression in Guatemala: sociodemographic characteristics and service access 

Dear Dr. Naber:

I'm pleased to inform you that your manuscript has been deemed suitable for publication in PLOS ONE. Congratulations! Your manuscript is now with our production department. 

Kind regards, 

on behalf of

Prof. Seo Ah Hong 

Academic Editor

PLOS ONE